# AGENTS HELP AGENTS: EXPLORING TRAINING-FREE KNOWLEDGE DISTILLATION FOR SLMS IN DATA SCIENCE CODE GENERATION

## ABSTRACT

Knowledge distillation from Large Language Models (LLMs) to locally hosted Small Language Models (SLMs) provides advantages for Data Science Code Generation (DSCG) such as enhanced data privacy and reduced response times. However, achieving effective distillation without resource-intensive training is challenging. This paper investigates whether LLMs can distill knowledge to SLMs through In-Context Learning (ICL), a training-free method for rapid task adaptation. We present the **Agents Help Agents (AHA)** framework, which facilitates automatic knowledge distillation from LLMs to SLMs via agent orchestration. AHA consists of three phases: exploration through an **Agent Orchestration Interface (AOI)**, memory collection of successful examples, and inference augmented with distilled knowledge. The AOI orchestrates interactions between a LLM as a Teacher Agent and a SLM as a Student Agent. And we propose two distillation strategies: a static approach that aggregates an offline instruction set and a dynamic RAG-based approach that distills knowledge dynamically during inference. We evaluate AHA on three challenging code generation tasks for tabular data analysis: TABMWP, BIRD-SQL, and WIKITQ. Experimental results demonstrate the effectiveness of AHA, leading to an average 27.5% relative improvement in the performance of the Student Agent **Phi-3-mini**. Additionally, relative gains of 14.3% and 30.9% are observed in **Llama-3.1-8B** and **GPT-35-Turbo**, respectively, even though those models were not calibrated as part of the orchestration, highlighting the model-agnostic nature of the distilled knowledge in AHA. Further analysis compares distillation and demonstration techniques across different data input settings, providing insights into optimal configurations for DSCG.

## 1 INTRODUCTION

Data Science Code Generation (DSCG) automates the conversion of natural language queries into executable code, empowering non-expert information extraction and analysis from tabular data efficiently. This process enhances productivity, reduces the technical barrier for data analysis, and allows data scientists to focus on deriving insights, ultimately supporting more effective decision-making (Khanbabaei et al., 2018; Han et al., 2011; Fayyad et al., 1996). This is a challenging task since it not only requires code generation capability but also data understanding capability.

Large Language Models (LLMs) have demonstrated remarkable performance across diverse, complex tasks (Singh et al., 2023; Mu et al., 2024; Chen et al., 2023; Zheng et al., 2024; Deng et al., 2023). Leveraging LLMs or LLM agents for automatic code generation from user queries offers an effective solution (Yang et al., 2024b; Wang et al., 2024b). However, the integration of LLMs in DSCG faces two primary challenges: 1) Privacy concerns arise when utilizing closed-source LLMs such as GPT-4 (Achiam et al., 2023) or Claude-3.5-Sonnet (Ogunseyi et al., 2023). 2) Deploying large open-source models like Llama-3.1-405B (Dubey et al., 2024) or DeepSeek-v2 (236B) (Liu et al., 2024) can be challenging due to their large number of parameters. Balancing these benefits and challenges is crucial for effective data science applications.

Small Language Models (SLMs), such as Phi-3-mini (Abdin et al., 2024) and Llama-3.1-8B (Dubey et al., 2024), have gained attention for their In-Context Learning (ICL) capabilities but more advantages for local deployment and on-device inference. These models offer computational efficiency and enhanced data privacy, crucial for resource-constrained or privacy-sensitive applications (Joshi et al., 2024). While SLMs have shown competitive performance in some general tasks including natural language understanding (Nie et al., 2020) and code completion (Chen et al., 2021), their effectiveness in data science code generation tasks remains an open question.

Fine-tuning is a common strategy to enhance SLM capabilities for complex tasks (Petroni et al., 2021). However, this approach encounters several challenges in the domain of data science DSCG. One primary issue is the limited availability of high-quality training data. Professional tabular datasets, such as relational databases, are often small or proprietary, restricting access to substantial corpora for training. Additionally, the dual expertise required in both coding syntax and data understanding for accurate annotation further constrains dataset scalability (Li et al., 2024b; Lei et al., 2024). This is reflected in recent benchmarks for data science code generation, which typically contain around or fewer than 1,000 samples, highlighting the complexity and resource constraints in this field (Hu et al., 2024; Agashe et al., 2019; Lai et al., 2023; Zhang et al., 2023; Yin et al., 2023). Recent research has explored distillation from LLMs to SLMs through fine-tuning on synthetic data generated (Team et al., 2024; Magister et al., 2023; Kang et al., 2023). While this approach shows promise, several challenges persist. For example, frequent updates to code packages introduce new syntax that may conflict with previously trained knowledge (Wu et al., 2024). Furthermore, the performance improvements obtained from these methods often fail to generalize well across different programming languages or frameworks, requiring extensive re-training for each package update or new task (Shen et al., 2022a; Ke et al., 2023). However, In-Context Learning (ICL) can adapt to new requirements or tasks by providing relevant instructions or examples, reducing the effort required for re-training or continual training. This raises a research question in DS code generation: ***Can LLMs distill knowledge to SLMs through In-Context Learning (ICL)?***

In this paper, we explore the potential of knowledge distillation from LLMs to SLMs via ICL. We design Agents Help Agents (AHA), a novel, fully automated framework that enables LLM as a Teacher Agent to guide SLMs as Student Agents in complex data science code generation tasks. AHA operates in three phases: **exploration**, **memory database collection**, and **knowledge-driven inference**. During exploration, we employ the Agent Orchestration Interface (**AOI**) that allows an LLM to probe and analyze SLM code knowledge by converting questions into step-wise functional plans and asking SLMs to infill the code for each plan. Then, successful collaborated cases are stored in memory databases. We also introduce two novel distillation techniques during inference: General Instructions and Meta Instructions.

We evaluate AHA on three challenging tabular analysis datasets that need code generation: TABMWP (Lu et al., 2023), BIRD-SQL (Li et al., 2024a), and WIKITQ (Pasupat & Liang, 2015). The experimental results demonstrate that the AHA framework significantly improves the performance of SLMs across all datasets, validating the potential of our knowledge distillation approach via In-Context Learning (ICL). Notably, the performance boost achieved by AHA is not limited to the specific SLM trained during the orchestration process but generalizes across other SLMs as well. This model-agnostic nature further highlights the flexibility and adaptability of distilled knowledge in our framework, enabling it to be applied in a wide range of data science code generation scenarios without requiring extensive retraining.

## 2 METHODOLOGY

### 2.1 TASK FORMULATION

Given a natural language query or question $q_i \in \mathcal{Q}$, where $\mathcal{Q} = \{q_1, q_2, \ldots, q_N\}$ represents a set of $N$ queries or instructions, and its corresponding tabular data or database schema information $d_i \in \mathcal{D}$, where $\mathcal{D} = \{d_1, d_2, \ldots, d_N\}$, the Small Language Model (SLM) is tasked with generating a single, concise, and executable code snippet $c_i$. This code snippet must accurately answer the query $q_i$ with the associated data $d_i$. The function that maps each query-data pair to its corresponding code snippet by SLMs is defined as $f_{\text{gen}}$, and can be written as:

$$c_i = f_{\text{gen}}(q_i, d_i) \quad \text{for } i = 1, 2, \ldots, N. \tag{1}$$

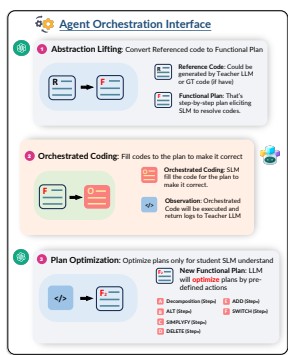 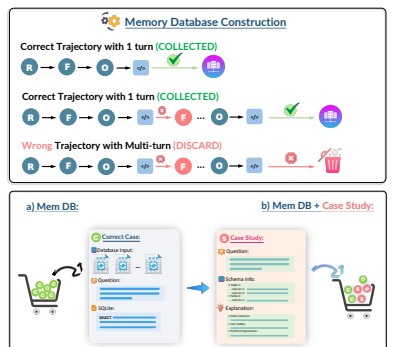 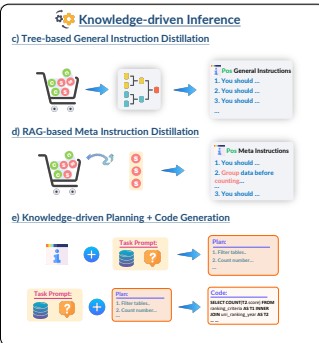

Figure 1: Overview of the AI agent orchestration system for data science code generation. Left: Agent Orchestration Interface (AOI) with abstraction lifting, orchestrated coding, and plan optimization. Center: Memory Database Construction, including trajectory collection and case study integration. Right: Knowledge-driven Inference and Planning, featuring tree-based general instruction distillation, RAG-based meta instruction generation, and knowledge-driven code generation.

## 2.2 AGENT ORCHESTRATION INTERFACE

The Agent Orchestration Interface (AOI) is designed to mediate the interaction between a Large Language Model (LLM), a Teacher Agent, and a Student Agent, represented by the SLM. The primary goal of the AOI is to generate successful and informative problem-solving cases, which are later used for knowledge distillation in DSCG. The AOI is composed of three key components: Abstraction Lifting, Orchestrated Coding, and Plan Optimization.

**Abstraction Lifting (AL).** In this phase, LLM generates a functional plan $P_i = \{s_{i1}, s_{i2}, \ldots, s_{iK}\}$ based on a query $q_i$, data input $d_i$, and the corresponding ground truth (gt) code $\tilde{c}_i$. The ground truth code $\tilde{c}_i$ can either be sourced from an existing dataset or generated by the Teacher Agent when it is not directly available but a ground-truth answer string exists. This functional plan is defined as $\mathcal{L}_{al}(d_i, q_i, \tilde{c}_i)$, where $\mathcal{L}_{al}$ LLM performing abstraction lifting. Each step $s_{ij}$ in the plan corresponds to a key subtask derived from the query, collectively forming a structured template outlining the solution process. These steps are annotated by the LLM with descriptive comments and placeholders such as `[Fill Your Code Here]` in Python or `[Fill Your Sub-Query]` in SQL, as shown in Figure 2, ensuring that the SLM follows the logical flow of the entire plan and enables guided code generation. Unlike Chain-Of-Thought (Wei et al., 2022) plans, which provide intermediate steps in continuous textual form, our approach bridges high-level problem understanding with low-level code implementation logic, allowing the SLM to follow the better plan for data science code generation.

**Orchestrated Coding (OC).** Once the functional plan $P_i$ is provided, the SLM considers all context including the question and data input to generate the complete orchestrated code $c_i = f_{gen}(d_i, q_i, P_i)$ in a single turn by filling all placeholders, ensuring the solution is correct and executable. The results from executing this orchestrated code are then compared to those from a reference solution (such as ground truth answer string or gt codes) to evaluate whether the SLM fully understands both the data and the logic needed to answer the question. This comparison serves as a key indicator of the problem-solving accuracy of SLM and alignment with the intended solution. While the ground truth code may already be available from datasets or generated by the Teacher Agent, orchestrated coding and abstraction lifting are crucial for a few reasons. First, AL breaks down complex problem-solving tasks into manageable sub-tasks, with the potential to improve the performance of the SLMs across a wide range of analytical queries by assisting them in understanding modular structure. Additionally, error isolation can be grounded in the program structure, enabling more precise identification of issues and contributing to optimized plans. This is supported by our analysis in Section 3.6 that compares chain-of-thought with functional plans across multiple turns of orchestration.

**Plan Optimization (PO).** The plan optimization process is an iterative procedure that unfolds over multiple turns, denoted by $t$. During each iteration, the SLM refines the functional plan $P_i^t$. To formalize this interactive optimization process, we define an environment $\mathcal{E} = \langle \mathcal{S}, \mathcal{A}, \mathcal{O}, \mathcal{T} \rangle$, following (Zhou et al., 2023; Xie et al., 2024; Gu et al., 2024), where $\mathcal{S}$ represents the state space,

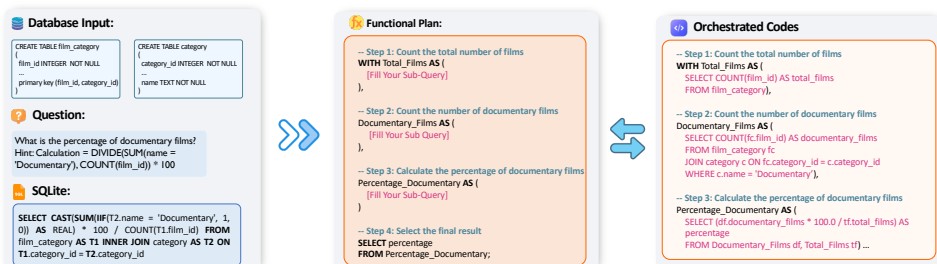

Figure 2: Main steps of AOI demonstrated with a text-to-SQL task example. Teacher Agent converts referenced code to a functional plan for Student Agent to complete. Teacher Agent iteratively optimizes the plan until the Student Agent produces correct code. A Python code AOI example is provided in Appendix D.1.

$\mathcal{A}$ the action space (see Table 1), and $\mathcal{O}$ the observation space. In this context, the plan $P_i^t$ is embedded within the current state $\mathcal{S}_i^t$, serving as a structure that guides the SLM to generate code. The orchestrated code $c_i^t$ is the snippet produced by performing the plan $P_i^t$ within the environment.

During each turn, the LLM observes $o_i^t$, the outcome of executing orchestrated code $c_i^t$ generated by the SLM, and selects an action $a_i^t$ from $\mathcal{A}$ to optimize the plan. For example, if a step $s_{ij}^t$ contains an error such as: "Step j: List players who was born before 1930 and after 1950", the LLM will apply an action ALT($\cdot$) to correct this, resulting in an updated plan $P_i^{t+1}$ with the refined step $s_{ij}^{t+1}$ as "Step j: List players who were born after 1930 and before 1950". This iterative process can be represented as:

$$P_i^{t+1} = \{s_{i1}^t, s_{i2}^t, \ldots, s_{ij}^{t+1}, \ldots, s_{iK}^t\}, \quad s_{ij}^{t+1} = \mathcal{L}_{\text{opt}}(s_{ij}^t, o_i^t, a_i^t). \tag{2}$$

Here, $s_{ij}^{t+1}$ is updated by the optimization function $\mathcal{L}_{\text{opt}}$ of the LLM, which integrates the current observation $o_i^t$, action $a_i^t$ and sub-optimal step $s_{ij}^t$. The system transitions from state $\mathcal{S}_i^t$ to $\mathcal{S}_i^{t+1}$ through $\mathcal{T}(\mathcal{S}_i^t, \mathcal{A}_i^t)$, resulting in the updated plan $P_i^{t+1}$. The SLM then generates the updated orchestrated code $c_i^{t+1} = f_{\text{gen}}(q_i, d_i, P_i^{t+1})$ for the new plan. This process repeats until the output is correct or the maximum number of iterations $T$ is reached.

## 2.3 MEMORY DATABASE CONSTRUCTION

After interactions between the LLM and SLM in AOI, the finalized states are stored in a memory database. This database includes the correct orchestrated codes, along with the context of the question and data input. This process ensures that the SLM can efficiently reference and apply related knowledge to new, unseen queries.

**Case Study Translation.** Rather than only storing raw, heterogeneous cases that consist of a query, plan, and orchestrated code in a simple stacked format, the LLM refines these into case study-like representations. These representations distill the reasoning behind the success of each example, serving as an intermediate abstraction that emphasizes the underlying rationale for the chosen approach. Each case study $G_i$ contains a Case Name, Question, Schema / Value Information, Objective, and an Explanation of how the solution code successfully addresses the query using the provided data. An example of this structure is provided in Appendix I. As shown in Figure 1 (Mid), AHA performs case study translation only for correct cases, because reflecting on incorrect cases without supervision or comparison to correct cases often introduces hallucinations.

**Correct Case Collection.** The Correct Case Collection, denoted as $\mathcal{M}$, consists of cases where the SLM has generated correct orchestrated codes. Each case $M_i$ in this collection contains the natural language query $q_i$, the corresponding data $d_i$, the correct orchestrated code $\hat{c}_i$, which contains descriptive comments as shown in Figure 2 (right), and the case study $G_i$ illustrating the solution. The set $\mathcal{M}$ is the union of all such individual cases:

$$\mathcal{M} = \bigcup_i M_i, \quad M_i = (q_i, d_i, \hat{c}_i, G_i). \tag{3}$$

| Action Type | Expression | Description |
|---|---|---|
| **Decomposition** | `step(x) → step(a), step(b)` | Split a complex step `x` into smaller, manageable steps such as step `a` and step `b`. |
| **ALT** | `step(x) → step(y)` | Replace a step `x` described by ambiguous or incorrect messages with a clearer and correct alternative step `y`. |
| **ADD** | `step(x) → step(x), step(a)` | Add a necessary step `a` to ensure the completeness of code logic. |
| **DELETE** | `step(x) → None` | Remove the unnecessary step `x`, which may lead to misunderstanding by the SLM. |
| **SIMPLIFY** | `step(x) → simple_step(x)` | Replace a complex step `x` with a simpler approach. For example, convert recursive plans into iterative loops. |
| **SWITCH** | `packageA.step(x) → packageB.step(x)` | Use a simpler package to achieve the same functionality. For example, conversion from Package `Linear Regression` to `Correlation Coefficient` to determine relationship between two variables. |

Table 1: The 6 action types utilized by the LLM during the Agent Orchestration Interface (AOI) to optimize the plans for better understanding and code generation by SLMs

## 2.4 KNOWLEDGE DISTILLATION FROM MEMORY DATABASE

This part presents two methods for distilling knowledge from the memory database: fine-to-coarse general instruction generation and RAG-based meta-instruction generation. Distilled knowledge then guides the SLM in learning how to plan and generate code more accurately for unseen queries.

**Fine-to-Coarse Knowledge Distillation for General Instructions Generation.** Our first approach generates universally applicable, "plug-and-play" instructions through a novel fine-to-coarse knowledge distillation method. This technique employs a recursive, tree-based strategy, offering an alternative to conventional sequential updating methods. Approaches such as (Askari et al., 2024) process batches of examples sequentially and need to select initial examples that often require human efforts. In contrast, our method aims to enable a fully automated workflow, constructing a knowledge tree recursively and in parallel, thereby reducing both biases and the need for human intervention. We define the set of distilled **General Instruction** $\mathcal{I}_g$ as:

$$\mathcal{I}_g = \mathcal{L}_{\text{sum}}(\mathcal{M}) = T_l, \quad T_l = \mathcal{L}_{\text{agg}}(T_{l-1}), \tag{4}$$

where $\mathcal{L}_{\text{sum}}$ is a recursive function executed by the LLM, and $T_l$ is the root node of the tree with the highest layer $l$, which encapsulates task-specific knowledge, enabling the SLMs to apply it to new examples.

At each recursive step, nodes at layer $l$ aggregate knowledge from layer $l - 1$ using the aggregation function $\mathcal{L}_{\text{agg}}$, which summarizes local instructions within each batch (see prompts in Figure 12, 13 of Appendix H for reference). The leaves represent the individual case studies recorded in $\mathcal{M}$, while higher layers abstract and generalize knowledge from lower ones. The layer of the tree and the final number of distilled rules are hyper-parameters, allowing for balancing complexity and generalization capability. This hierarchical structure can capture broadly applicable rules, resulting in a more general instruction. General Instruction then guides the SLM in generating correct code for unseen queries in a plug-and-play manner represented by Figure 1 (Right). A detailed illustration with examples can be found in Appendix A.

**RAG-based Knowledge Distillation for Meta Instruction Generation.** General Instructions, while broadly applicable, often fall short when addressing long-tail problems. To mitigate this gap, we propose a Retrieval-Augmented Generation (RAG) framework for localized instruction distillation. In the retrieval phase, we identify the top relevant examples from a memory database $\mathcal{M}$ via an embedding model. Relevance is measured via a function $\mathcal{D}$, expressed as $\mathcal{R}(q_i) = \mathcal{D}(q_i, \mathcal{M}, k)$, where $k$ is number of most relevant cases.

Then, rather than leveraging the entire heterogeneous context, the case studies of these retrieved examples are then fed into the SLM to extrapolate plans for solutions, adhering them to the specific query. Here, SLM performs a secondary distillation, extracting shared knowledge patterns from these case studies, which have already been distilled by the LLM (Teacher Agent), to generate instructions, noted as **Meta-Instruction ($\mathcal{I}_{\mathbf{m}}(q_i)$)**, precisely specific to the current query at hand. The process is formalized:

$$\mathcal{I}_{\mathrm{m}}(q_i) = f_{\mathrm{agg}}(q_i, \mathcal{R}(q_i)) \tag{5}$$

where $f_{\mathrm{agg}}$ is an aggregation function applied by the SLM. By doing so, SLMs can generate more relevant and contextually appropriate instructions, effectively bridging the gap between general knowledge and query-specific requirements.

**Knowledge-Driven Inference.** Harnessing the distilled instructions $\mathcal{I} \in (\mathcal{I}g, \mathcal{I}m(q_i))$ from the memory database, the SLM initially formulates a structured plan $p_{\mathrm{gen}}$, which it subsequently employs to generate code for new queries. For a given query $q_i$ and its associated data $d_i$, this process unfolds as follows:

$$P_{\mathrm{gen}} = f_{\mathrm{plan}}(q_i, d_i, \mathcal{I}), \quad c_i = f_{\mathrm{gen}}(P_{\mathrm{gen}}, q_i, d_i), \tag{6}$$

where $f_{\mathrm{plan}}$ denotes the planning function executed by the SLM. This plan serves as a blueprint, guiding the following code generation phase. The SLM then employs the function $f_{\mathrm{gen}}$, which takes $P_{\mathrm{gen}}$ along with the original query $q_i$ and data $d_i$ to generate the final code $c_i$.

## 3 EXPERIMENTS

### 3.1 DATASETS AND METRICS

We evaluate our approach on three tabular data analysis datasets: WIKITQ (Pasupat & Liang, 2015), TABMWP (Lu et al., 2023), and BIRD-SQL (Li et al., 2024a). These datasets cover various task types and data complexities, challenging models to interpret different data structures and generate accurate, executable code for question answering. The data statistics are shown in the Table 2. 1). Questions in WIKITQ typically involve operations such as counting, comparison, and aggregation (e.g., `How many players scored more than 10 points?`, `What is the largest city by population?`).
We sample 1,000 instances from the test set ($\sim 25\%$ of the full set) and 2,000 examples from the training set for exploration. Performance is evaluated using the accuracy metric as implemented by the official evaluation

| STATISTIC | TABMWP | WIKITQ | BIRD-SQL |
|---|---|---|---|
| **Dataset Features** | | | |
| ⊘ # train examples | 1,000 | 2,000 | 1,000 |
| ⊘ # eval examples | 1,000 | 1,000 | 500 |
| ⊘ question type | Analysis | SP | SP + Analysis |
| ⊘ # toks / Q | 26.5 | 12.6 | 20.0 |
| **Data Structure** | | | |
| ⊘ data input type | Single | Single | RDB |
| ⊘ # rows / data | 6.13 | 28.5 | 354k |
| ⊘ # columns / data | 2.22 | 6.36 | 73.3 |
| **Code Features** | | | |
| ⊘ code type | Python | Python | SQL |
| ⊘ answer type | String | String | Code |
| ⊘ # toks / code | N/A | N/A | 61.15 |

Table 2: Statistics for three datasets. The term `Analysis` indicates that the dataset mainly consists of analytical questions, while `SP` refers to semantic parsing tasks.

scripts (Pasupat & Liang, 2015), which measures the correctness of the final answer derived from the generated codes. 2). In TABMWP, questions focus on mathematical word problems involving tabular data, extending beyond semantic parsing to include data analysis questions (e.g., `Is there a relationship between x and y?`). We use 1,000 instances from the development set for memory construction and evaluate on the full test set (1,000 non-overlapping questions). Performance is evaluated by comparing the generated results with the ground truth answers across all grade levels. 3). **BIRD-SQL** presents the most complex data structures and comprehensive question types in our evaluation. The data inputs are relational databases, which are more challenging than the single tables in WIKITQ and TABMWP. The questions contain both semantic parsing and analytical tasks. For exploration, we adopt the mini-train set curated by (Qu et al., 2024), which comprises 1,000 examples. Our evaluation is conducted on the mini-dev set, a collection of 500 high-quality and challenging cases officially selected by the BIRD-SQL team. We evaluate performance by widely adopted execution accuracy (EX) metric for this dataset.

| Model | WIKITQ | TABMWP | | | BIRD-SQL | | | |
|---|---|---|---|---|---|---|---|---|
| | Accuracy | Grad. 1-6 | Grad. 7-8 | Total | Sim. | Med. | Chal. | Total |
| CodeLlama-7B | 11.80 | 26.55 | 13.11 | 20.50 | 43.92 | 18.00 | 11.76 | 24.40 |
| CodeLlama-13B | 34.90 | 37.27 | 24.22 | 31.40 | 45.27 | 17.65 | 17.65 | 26.80 |
| StarCoder2-7B | 20.70 | 34.00 | 27.56 | 31.10 | 41.22 | 21.60 | 17.65 | 26.60 |
| StarCoder2-15B | 36.60 | 39.09 | 36.44 | 37.90 | 55.41 | 30.40 | 14.71 | 34.60 |
| Phi-3-Small-7B | 27.00 | 46.36 | 38.00 | 42.60 | 52.03 | 28.40 | 10.78 | 31.80 |
| Phi-3-Medium-14B | 44.80 | 59.45 | 46.00 | 53.40 | 51.35 | 32.80 | 13.73 | 34.40 |
| *Student Agent Performance* | | | | | | | | |
| **Phi-3-Mini-3.8B** | 32.50 | 44.18 | 38.89 | 41.80 | 38.51 | 21.20 | 11.76 | 24.40 |
| + Chain-Of-Thought | 27.70 | 46.36 | 35.33 | 41.40 | 34.46 | 22.00 | 12.75 | 23.80 |
| + Static Few-Shot | 23.00 | 37.27 | 34.89 | 36.20 | 47.97 | 20.80 | 7.84 | 26.20 |
| + Dynamic Few-Shot | 16.60 | **51.45** | **52.89** | **52.10** | 42.57 | 18.80 | 11.76 | 24.40 |
| + AHA General Instruction (Ours) | 39.50 | 50.91 | 46.89 | 49.10 | **51.35** | 25.60 | **17.65** | 31.60 |
| + AHA Meta Instruction (Ours) | **41.10** | 48.36 | 42.44 | 45.70 | **51.35** | 30.40 | 16.67 | **33.80** |

Table 3: Performance comparison of various SLMs on WIKITQ, TABMWP, and BIRD-SQL, with results presented in accuracy percentages. Improvements of our AHA methods over the End-to-End Code Gen baseline are highlighted using different intensities of olive color. **Bold** indicates best results for Phi-3-Mini, while underlines denote second-best results.

## 3.2 IMPLEMENTATIONS

**Setup.** Experiments are conducted on three datasets across two primary settings. For TABMWP and WIKITQ, the SLMs are instructed to generate Python Pandas code to answer questions. However, since TABMWP and WIKITQ are QA datasets lacking ground truth code, two additional steps are implemented. First, the Teacher Agent is employed to generate initial Python code solutions as referenced or ground-truth code $\tilde{c}_i$, prior to the Agent Orchestration Interface (AOI). Second, following both orchestrated and inference code generation, the SLM is called upon to produce concise string answers for final accuracy evaluation. This process involved an additional step: given the question $q_i$ and executed results $o_i$, the SLM generated a result string $r_i = f_{\text{ans}}(q_i, o_i)$ with an answering prompt, which was then compared to the GT answer string using the official evaluation script. For BIRD-SQL, a SQLite environment is established for orchestration and evaluation, following the task formulation $c_i = f_{\text{gen}}(q_i, d_i)$. We do not need to generate initial SQL by LLMs as ground truth codes since they already contain ground truth SQLs.

For General Instruction generation, we set layer of the tree $l = 2$ and limit number of rules to under 10. In RAG-based Meta Instruction generation, we employ KNN with L2 distance, setting $k = 3$ for top relevant cases and using CodeT5+ (Wang et al., 2023d) as the embedding model. Details are in Appendix C.

**Baselines Models and Methods.** We define an SLM as suitable for this task if it satisfies two criteria: (1) it can perform reasoning through in-context learning (ICL) without relying solely on fine-tuning, and (2) it has fewer than 15 billion parameters ($< 15B$), enabling inference on an A100 GPU or less powerful hardware. For closed-source models, we choose GPT-35-Turbo as SLM since it has faster inference speed and its performance falls behind other larger models such as GPT-4-Turbo or GPT-4o. We implement models for three purposes: 1) **Orchestration Models**: In our experiment, we select LLM GPT-4o (Achiam et al., 2023) as Teacher Agent and a SLM Phi-3-mini-128k (Abdin et al., 2024) as Student Agent, which only contains $< 3.8B$ parameters. 2) **Evaluation Models**: There are several families of SLMs for evaluation. Phi-3 models (Abdin et al., 2024), CodeLlama models (Roziere et al., 2023), StarCoder2 family (Lozhkov et al., 2024). 3) **Knowledge-Transmission Models**: We include the widely-used closed-source model GPT-35-Turbo and Llama-3.1-8B as new models for evaluation of knowledge transmissions in Section 3.5. Our focus in this paper is on single-pass code generation. Thus, the environment is not available for SLMs to iteratively refine or generate code in multiple turns as in (Yao et al., 2023; Wang et al., 2024c). We consider zero-shot end-to-end code generation, Chain-Of-Thought (Wei et al., 2022), Static Few-shot Demonstration (Brown et al., 2020), Dynamic RAG-based Few-shot Demonstrations (Gao et al., 2024) as our baseline methods. For fairness, we employ three examples for all few-shot demonstration methods.

## 3.3 OVERALL RESULTS

**Overall Performance.** Table 3 highlights three key aspects: (1) Knowledge distilled from **AHA** can make Phi-3-mini outperform both the End-to-End Code Generation baseline and the widely-

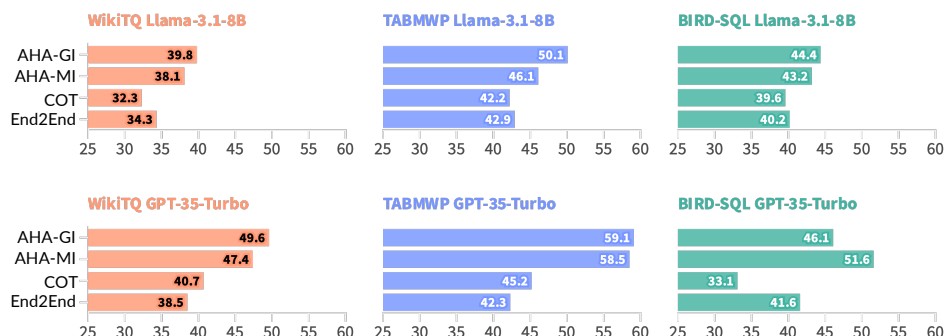

Figure 3: Knowledge transmission from the memory database between GPT-4o and Phi-3-mini across three datasets. The results demonstrate that knowledge distilled from AHA can transfer to new models.

used Chain-of-Thought reasoning technique across all datasets for SLM. Specifically, the Phi-3-mini model demonstrates relative improvements ranging from **17.5%** on TABMWP to **38.5%** on BIRD-SQL. (2) The enhanced Phi-3-Mini frequently matches or exceeds the performance of larger models, especially those with 2-3 times more parameters, notably surpassing CodeLlama-13B by **17.7%**, StarCoder2-15B by **11.2%** on the TABMWP benchmark and approaching the performance of Phi-3-Medium (which has 4x times the parameters) across all datasets. (3) Our experiments also indicate that Chain-of-Thought reasoning can negatively impact SLM performance. In such complex scenarios, we observe that SLMs often generate hallucinations, resulting in incorrect reasoning steps. The propagation of these errors due to flawed or invalid thought processes ultimately leads to diminished performance (Yee et al., 2024).

### 3.4 DISTILLATION V.S. DEMONSTRATION

In this section, given the memory database, we compare the effectiveness of our knowledge distillation techniques, with conventional demonstration-based strategies. In our approach, distillation involves transferring knowledge from the memory database to SLMs through task-specific instructions. On the other hand, demonstration-based methods guide SLMs by presenting explicit task examples to facilitate analog reasoning (Yu et al., 2024). We implemented two variants of few-shot demonstrations: **Static**: Human experts select three representative examples from the memory database, which remain constant across all cases. **Dynamic RAG-based**: Examples are selected from AHA memory database based on similarity to the current query. For fair comparison, we also implement the same RAG system as AHA-MI, described in Section 3.2.

Our findings indicate that few-shot demonstration generally underperforms AHA knowledge distillation technqiues on each dataset. However, we observe a surprisingly superior performance of the RAG-based few-shot demonstration compared to our designed knowledge distillation and other baselines on TABMWP. This effectiveness appears to correlate with the complexity of the input data by further analysis. Referring to Table 2, we note that TABMWP presents the simplest data input, containing only **2.22** columns and **6.13** rows per data point, with clean values consisting of numbers or processed strings. However, when dealing with WIKITQ, which contains irregular value types, column names, and BIRD-SQL, which presents complex database schemas and values, SLMs exhibit confusion with such heterogeneous and complex inputs. More critically, SLMs generate 38.2% more invalid outputs (e.g., `"SELECT \n\n\n\n..."`) in BIRD-SQL.

Based on these observations, we conclude that dynamic few-shot demonstration is more convenient and effective for leveraging the memory database when the input data is less complex. On the contrary, for complex data such as tables with dirty values or relational databases, our designed knowledge distillation enables SLMs to better utilize knowledge and perform tasks more effectively. It is worth noting that in real-world scenarios, complex data schemas and inputs are prevalent (Lee et al., 2021). Moreover, our approach exhibits greater scalability as task complexity increases. Although dynamic few-shot learning achieves a slight 3.0% advantage over our method on simpler tasks, our technique outperforms it by a significant 16.2% on more complex systems. This asymmetry in performance gains highlights the robust generalization of our knowledge distillation approach

for DSCG tasks across a spectrum of input complexities, from simple to more challenging data inputs.

### 3.5 KNOWLEDGE TRANSMISSION

While AHA shows notable performance gains for SLMs in data science code generation without fine-tuning, an important question arises: ***Is the distilled knowledge only useful to the Student Agent participated in Orchestration?*** In order to answer this, we conduct knowledge transmission experiments by `Llama-3.1-8B` and `GPT-35-Turbo`, which didn't attend the exploration.

The results in Figure 3 demonstrate that both General and Meta Instructions, distilled from AOI memory database between GPT-4o and Phi-3-mini, obviously benefit these new models. AHA-GI and AHA-MI consistently outperform conventional techniques like COT and End2End across all datasets leading to average relative improvement of 14.3% for Llama-3.1-8B and 30.9% for GPT-35-Turbo. This proves that distilled knowledge is not limited to the original Student Agent (Phi-3-mini) but can transfer effectively to other models without additional fine-tuning, suggesting an efficient pathway for knowledge augmentation in emerging SLMs.

### 3.6 ORCHESTRATION MEDIA TYPE ANALYSIS

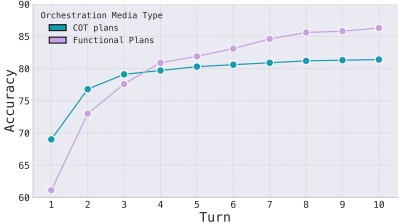

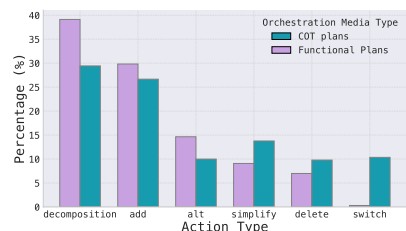

(a) Accuracy across different AOI turns when using COT and Functional Plans as orchestration media.

(b) Action type distribution for plan optimization in AOI using two orchestration media types.

Figure 4: Comparison of accuracy and action type distribution for orchestration media types in AOI. The experiments are conducted on TABMWP on 1000 training examples across 10 turns.

In this section, we assess the impact of various orchestration media types on data efficiency within the AOI frameworkduring exploration. Figure 4 (a) presents a comparison of the Phi-3-mini performance growth trends adopting COT plans, the sequential textual plan, versus functional plans over 10 turns of plan optimization on 1000 training data in AOI.

At the beginning, COT plans enable Phi-3-mini to outperform the functional plans (69.00% vs. 61.10%). However, as orchestration continues, the functional plans progressively improve, eventually surpassing the COT plans, achieving 86.3% compared to 81.4% by the final round. A visualization of action distributions for plan optimization, performed by GPT-4o in Figure 4 (b), indicates that `Decomposition` and `Add` occur much more frequently than other actions, generating longer plans with more steps and interpretations. In such scenario, Phi-3-mini demonstrates significant hallucinations when processing extended COT plans, especially when the number of steps exceeds 7, we observe that Phi-3-mini would ignore some steps of the plan and hallucinate some steps that do not appear in the orignal plan. In contrast, the structured nature of our functional plan forces Phi-3-mini to follow each step methodically, ensuring the completion of all placeholders. This structured approach provides a clearer sense of task progression since the model perceives the task as completed only when all placeholders are filled.

In conclusion, functional plans lead to a larger portion of correct cases, promoting a more data-efficient strategy for constructing memory databases, which can be effectively leveraged by SLM agents. This finding can prove that our designed functional plans are better orchestration media type compared to general COT plans in data science code generation task.

### 4 RELATED WORK

**Data Science Code Generation (DSCG).** DSCG focuses on automating code generation for data-centric tasks, requiring a deep understanding of data formats like CSV, TSV, and relational databases

(RDB). Unlike general code generation models, which primarily generate syntactically correct code in response to natural language instructions (Chen et al., 2021; Luo et al., 2024), DSCG must ensure that the generated code correctly interacts with underlying data structures. This involves understanding the schema, format, and semantics of the data, whether in Python code for handling tabular data (Chen et al., 2024; Cheng et al., 2023; Shen et al., 2022b) or SQL for interacting with relational databases (Yu et al., 2018; Lee et al., 2021; Li et al., 2024a). Spreadsheet-based code generation further extends DSCG, automating the generation of formulas and operations in tools (Wang et al., 2023a; Bhatia et al., 2023). Even though large language models (LLMs) have demonstrated effectiveness in enhancing the capabilities of SLMs, concerns regarding data privacy in cloud environments have prompted a reevaluation of their deployment strategies.

**Knowledge Distillation.** Knowledge distillation can mitigate this problem by transferring LLM capabilities to smaller models, enabling efficient deployment in resource-constrained environments (Xu et al., 2024). The field has evolved from early work on softened output training (Hinton, 2015) to advanced techniques like task-specific fine-tuning (Sanh, 2019), zero-shot learning (Wang et al., 2023b), and instruction-following datasets (Wang et al., 2023c;b). Progressive distillation techniques, such as the Orca framework (Mukherjee et al., 2023), demonstrate the potential for guiding the development of efficient open-source models. Self-distillation approaches have explored autonomous training data generation (Wang et al., 2023c). Recent advancements have focused on improving the performance and privacy aspects of DSCG by knowledge distillation (Luo et al., 2024). At the same time, synthetic data has been leveraged to enhance the generalization of SQL generation across different schemas (Yang et al., 2024a). Even though these techniques are effective, most still require training efforts to transfer knowledge. Our AHA framework introduces agent-based distillation through in-context learning, eliminating the need for task-specific fine-tuning and improving scalability across models and tasks.

**Agent Memory.** Agent memory can improve the capability of LLM-based agents, particularly in tasks that require long-term context retention and continuous knowledge accumulation (Zhang et al., 2024). Traditionally, research has focused on teaching LLMs to reflect on and evolve from memory built through their own interactions, limiting knowledge transfer to the model performing the task (Shinn et al., 2023). For DSCG, memory plays a critical role in managing complex data formats, maintaining long-term context, and learning from iterative analysis processes. For instance, reGAL (Stengel-Eskin et al., 2024) introduces a memory mechanism that enables LLMs to reuse abstractions across program synthesis tasks by storing and recalling reusable subroutines, significantly improving code generation performance. Similarly, models like MAGIC (Askari et al., 2024) have demonstrated how memory can facilitate self-correction in data analysis code generation. In more complex software engineering contexts, frameworks like SWE-Agent (Yang et al., 2024b) and OpenDevin (Wang et al., 2024b) by codeAct (Wang et al., 2024a) extend the use of memory by considering complicated contexts such as entire code repositories and prior interactions, allowing agents to manage more intricate tasks like cross-file dependencies and repository-level refactoring. However, current agent memory systems typically rely on a single model, i.e., memory is constructed and knowledge is learned and leveraged exclusively by models like GPT-4, limiting knowledge transfer. Our work introduces distillation techniques that enable SLMs to leverage memory orchestrated by multiple models, including more capable GPT-4o. This approach allows SLMs to utilize richer, external knowledge for improved performance in knowledge-driven ICL, effectively bridging the gap of knowledge sharing between high-capacity models and more efficient SLMs.

## 5 CONCLUSION

In this paper, we presented Agents Help Agents (AHA), an automatic framework for efficient knowledge distillation from Large Language Models (LLMs) to Small Language Models (SLMs) in Data Science Code Generation (DSCG). AHA leverages In-Context Learning to enhance SLM performance without fine-tuning, using agent orchestration and memory-based distillation to improve task accuracy. Evaluations on three challenging tabular data analysis datasets, which requires code generation, show a 27.5% relative performance increase for Phi-3-mini and model-agnostic effectiveness, benefiting models like Llama-3.1-8B and GPT-35-Turbo even they did not participate in the orchestration. These results highlight the potential of AHA for developing intelligent applications with a focus on privacy and computational efficiency.

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

# A  DETAILED DESCRIPTION OF FINE-TO-COARSE KNOWLEDGE DISTILLATION

We introduce a novel fine-to-coarse knowledge distillation method employing a recursive, tree-based approach to generalize to unseen queries. This method improves upon traditional sequential updating techniques by constructing a knowledge tree recursively and in parallel, thereby reducing bias and ensuring a more robust distillation process.

Our knowledge tree consists of nodes, each containing a batch of successful cases from the set $\mathcal{M}$. This structure allows for concurrent summarization of essential rules from diverse examples. We define the set of distilled instructions $\mathcal{I}$ as:

$$\mathcal{I} = \mathcal{L}_{\text{sum}}(\mathcal{M}) = T_l,$$

where $\mathcal{L}_{\text{sum}}$ is a recursive function executed by the Large Language Model (LLM) to distill knowledge, and $T_l$ is the root node of the tree. This function constructs a multi-layered tree, with each layer aggregating knowledge from the preceding layer. The tree's depth adjusts dynamically based on the context length of case studies, ensuring optimal abstraction at each layer.

At each recursive step, nodes in the current layer $l$ aggregate knowledge from layer $l-1$:

$$T_l = \mathcal{L}_{\text{agg}}(T_{l-1}),$$

where $\mathcal{L}_{\text{agg}}$ is the aggregation function merging batches of successful cases into more abstract representations.

The leaves of the tree (layer $L$) contain the original cases from $\mathcal{M}$, represented as batches:

$$T_L = \{\mathcal{B}_1, \mathcal{B}_2, ..., \mathcal{B}_k\}$$

where each batch $\mathcal{B}_j$ is a set of successful cases:

$$\mathcal{B}_j = \{M_{j1}, M_{j2}, ..., M_{jn}\}$$

and each successful case $M_{ji}$ is defined as:

$$M_{ji} = (q_{ji}, d_{ji}, \hat{c}_{ji}, S_{ji})$$

Here, $q_{ji}$ is the $i^{th}$ natural language query of $j^{th}$ batch, $d_{ji}$ is the $i^{th}$ corresponding data of $j^{th}$ batch, $\hat{c}_{ji}$ is the $i^{th}$ orchestrated correct code of $j^{th}$ batch, and $S_{ji}$ is the $i^{th}$ case study of $j^{th}$ batch summarizing the solution.

Each higher layer in the tree abstracts and summarizes the knowledge from the level below, culminating in the root node $T_l$, which represents the final set of distilled instructions $\mathcal{I}$.

This recursive and parallel tree construction allows for simultaneous extraction of rules, significantly reducing dependence on the order or selection of initial examples. Each node encompasses multiple successful cases, facilitating the extraction of generalized instructions through the identification of common patterns and rules.

The process continues iteratively from the leaves to the root, resulting in comprehensive and unbiased distilled instructions $\mathcal{I}$. This framework provides a well-rounded guide for the SLM in generating correct code for unseen queries, effectively balancing knowledge complexity with SLM constraints.

Our method represents an advancement in knowledge distillation for language models, offering a robust approach to extracting generalizable knowledge from diverse examples and enhancing SLM performance on unseen queries.

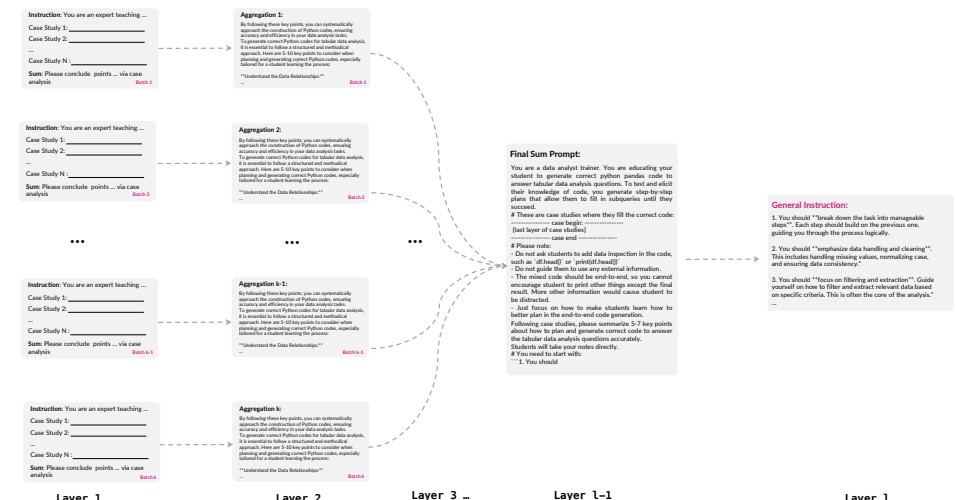

Figure 5: Illustration of how Fine-to-Coarse Knowledge Distillation for AHA-GI generation. The intermediate layers are omitted.

## B    MODEL IMPLEMENTATION

We implement models for three main categories of purpose:

### B.1    ORCHESTRATION MODELS

`gpt-4o`: The Teacher Agent (`gpt-4o`) is responsible for several key tasks, including Abstraction Lifting (see Section 2.2) and Plan Optimization (see Section 2.2), which are performed while monitoring the performance of the Student Agent. Additionally, the Teacher Agent handles the conversion of complex, heterogeneous cases into more readable case studies for Student Learning Models (SLMs), as detailed in Section 2.3. Finally, `gpt-4o` distills general instructions that contain task-specific knowledge, as described in Section 2.4. Notably, these general instructions are utilized by the SLMs in an offline manner, meaning that `gpt-4o` does not participate in the inference process of the SLMs.

`phi-3-mini-128k-instruct`: For the orchestration process, we select this 3.8B parameter SLM as the Student Agent due to its strong generalization abilities and efficient deployment.

### B.2    BASELINE MODELS

Within the orchestration mode, several families of Student Learning Models (SLMs) are evaluated. These include models from the Phi-3, Starcoder 2, and Llama families:

**Phi-3 Family**    (Abdin et al., 2024)

`phi-3-mini-128k-instruct` (3.8B)

`phi-3-small-128k-instruct` (7B)

`phi-3-medium-128k-instruct` (14B)

**Starcoder 2 Family**    (Lozhkov et al., 2024)

`starcoder2-7b-instruct`

`starcoder2-15b-instruct`

**Llama Family**    (Dubey et al., 2024)

```
codellama-7b-instruct-hf

codellama-13b-instruct-hf
```

### B.3 MODELS IN KNOWLEDGE TRANSMISSION

In Section 3.5, we explore the knowledge distilled from AHA to newly developed models, particularly in terms of their ability to generalize knowledge. For this evaluation, we select the following models:

`llama-3.1-8b-instruct`: This model is broad new, yet it shows significant performance improvements when leveraging the distilled knowledge.

`gpt-35-turbo-16k`: We also include a closed-source model in our experiments to demonstrate the effectiveness of our approach across both GPU-deployed and API-based models. Despite its number of parameters is unknown, we consider it as one of SLMs since its performance falls behind of its more advanced versions such as GPT-4.

## C DATASET IMPLEMENTATION DETAILS

### C.1 DATA FILE CONTENT

For convenient reproduction and following, we preprocess all dataset into more unified data format of `jsonl`. In python task (TABMWP, WIKITQ), each line of data contains `question_id`, `question`, `data_path`, `data_overview`, `answer_type`, `answer`. In SQL task (BIRD-SQL), each line of data contains `question_id`, `question`, `evidence`, `data_path`, `db_id`, `sql`.

### C.2 DATA INPUT CONTENT

The main goal of this work is to evaluate the code generation capabilities of models in understanding data schemas and structures across multiple datasets. Given the impracticality of providing all data values in real-world scenarios in which datasets may consist of millions or even billions of rows, we sample values for the part of data input to simulate realistic code generation tasks. We feed the markdown format of schemas with data samples as `data_overview`.

For **TABMWP**, we provide only the column names and the first three rows of values. This enables models to infer the data structure and value types necessary for Python Pandas code generation without exposing all the data.

For **WIKITQ**, which contains more complex and varied value types, we provide the first 10 rows of values and column names to help models navigate the dataset's intricacies.

In the case of BIRD-SQL, which contains relational databases with complex schemas and diverse value types, more advanced schema-linking techniques are often required to retrieve relevant tables or columns before answering queries (Wang et al., 2020; Pourreza & Rafiei, 2024). While we consider this advanced schema-linking process as future work for AHA, our current focus is on the code generation aspect. Therefore, we provide:

- Ground truth retrieved tables, reducing input complexity and simulating realistic human-machine interactions where users might supply potentially relevant tables.

- Full columns with column meaning description files.

- The first three rows of values for each table.

Although the retrieved tables are given, the models must still consider constraints and generate correct SQL queries. As shown in Table 3, performance on Bird-SQL remains relatively low, even with simplified table retrieval, highlighting the challenges of generating accurate SQL queries in complex database environments. This methodology allows us to evaluate code generation capabilities while approximating the real-world challenges of data analysis.

## C.3 PREVENT MODELS FROM DIRECT ANSWERING

We observe two kinds of direct answers behaviors in Python tasks, which leads to unfair evaluation of code generation ability:

**Unfair Data Inspection.** We find that SLMs usually generate data inspection codes such as `print(df.head())` in their code generation. When producing answers given executed results, such first few rows will show again in which SLMs tend to answer it correctly even with wrong codes or bugs. We need to decouple study of tabular understanding and code generation understanding data structure. And the mutually influenced capability would be the future goals of development.

**Data Leakage.** In datasets like WIKITQ, current popular SLMs tend to exhibit a form of data leakage, where models effectively **"memorize"** the ground truth answers, resulting in unfair evaluations. Through a sample of 100 generated codes using the Phi-3-mini baseline across these datasets, we observed that the model often embeds the correct answer directly into the code. For instance, given the question `"Who was the opponent of James V.?"`, the generated code might include a line like `opponent = "Smith W."`, which corresponds to the ground truth answer. The frequency of this leakage is particularly high in WIKITQ, where 23 out of 100 samples exhibits this behavior, especially for simpler question types. In contrast, datasets with more complex question structures, such as TABMWP, exhibits only 5 cases of leakage out of 100, while no instances are found in BIRD-SQL. These findings suggest that more complex input structures and question/code types can effectively reduce the possibility of data leakage.

**Mitigating Direct Answering Behaviors.** To address these cheating behaviors, we propose an **embodied prompt**, as outlined in Section H. This approach minimizes data leakage and prevents unfair data inspection during evaluation. As illustrated in Figure 7, we design scenarios where the model is informed that it has already inspected the dataset and does not need to generate further data inspection commands. Additionally, the embodied prompts encourage the model to approach tasks as a professional data analyst, preventing it from assigning variables based on memorized answers. Our evaluation shows that this method successfully eliminated data leakage in all 100 tested cases. Also, the unfair data inspections appear less frequently. The remaining of unfair data inspections will be removed by post-processed by regex functions.

We believe that this solution holds promise for addressing data leakage issues in complex benchmark evaluations. This is particularly important in DSCG, where datasets are difficult to collect from expert teams and frequent re-annotation to prevent leakage is impractical.

# D AHA FUNCTIONALITY

## D.1 AOI GENERALIZATION

Figure 6: Illustration of how AOI is conducted in Python for Tabular data analysis.

| Model | SIMPLE | MEDIUM | CHALLENGING | OVERALL |
|-------|--------|--------|-------------|---------|
| *Zero-Shot End-to-End Code Gen.* | | | | |
| Original Checkpoint | 38.51 | 21.20 | 11.76 | 24.40 |
| *LoRA Fine-Tuned* | 39.86 | 19.20 | 10.78 | 23.60 |
| | | | | |
| *AHA Knowledge Distillation* | | | | |
| General Instruction | **51.35** | 25.60 | **17.65** | 31.60 |
| Meta Instruction | **51.35** | **30.40** | 16.67 | **33.80** |

Table 4: Performance evaluation of Zero-Shot End-to-End Code Generation, LoRA fine-tuning, and our proposed knowledge distillation techniques on BIRD-SQL. Deeper red shading indicates a larger performance drop compared to the original pre-trained model, while green indicates no decline or improvement.

Our Agent Orchestration Interface (AOI) is adaptable to different programming languages with different data input settings. Figure 2 shows how AOI is conducted in RDB settings with SQLite, and Figure 6 shows how it's undertaken in Single-tabular data with Python.

## D.2 FINE-TUNING V.S. AHA KNOWLEDGE DISTILLATION

We also compare the performance of knowledge distillation via AHA with the commonly used LoRA fine-tuning method (Hu et al., 2022) under the same low-resource setting (1,000 training samples) on the BIRD-SQL dataset, specifically for the Phi-3-mini model. As shown in Table 4, training with such a limited amount of data can degrade the performance of SLMs. However, AHA significantly improves the performance of SLMs when utilizing the same data, with a clear margin of advantage. We hypothesize that: 1) the small training set may introduce bias, limiting the model generalization; and 2) LoRA fine-tuning struggles to teach SLMs the reasoning capabilities required for complex tasks within such an end-to-end training regime. On the Contrary, AHA leverages LLMs to automatically decompose difficult questions into more understandable steps to SLMs, and distill planning knowledge, which allows SLMs to generalize better when faced with new queries. In conclusion, AHA proves to be an effective method for enhancing the performance of SLMs in the domain of DSCG, which contains limited annotated data usually.

## E ABLATION STUDY

We conducted a comprehensive ablation study of AHA-MI, as shown in Table 5. Code-T5+ is a code embedding model (Wang et al., 2023d), while BGE-Large (Xiao et al., 2024) represents one of the state-of-the-art (SOTA) text embedding models. The study examines two types of RAG Index: one where distance is computed using question embeddings alone, and another where both question and schema embeddings are used. The "Plan + Gen" approach involves first constructing a plan with distilled knowledge, followed by generation using knowledge-driven planning. In contrast, the "Gen" approach involves direct generation without prior planning. The instruction type labeled `w/ examples` refers to cases where a specific example is provided by the Teacher Agent. We evaluate performance with 1, 3, and 5 examples to assess the impact of varying numbers of RAG examples. The results of the ablation study reveal several key insights:

**Code embeddings outperform text embeddings.** The superior performance of Code-T5+ over BGE-Large-en can be attributed to the nature of the task. While text embeddings emphasize on semantic and domain knowledge, code embeddings capture the syntactic and logical structure of coding problems, which is crucial for DSCG tasks. Even when presented with identical questions, the code solutions can vary significantly depending on the data input. Code-T5+ is able to effectively embed questions from a programming perspective, benefiting from its pre-trained corpus, whereas text embeddings are less suited for the task.

**Embedding only the question is more effective than embedding both the question and schema.** The study demonstrates that question-only embeddings lead to better results. This suggests that the inclusion of schema in the embedding may introduce unnecessary complexity, which may hinder performance on the DSCG task.

**Planning is essential for more complex tasks.** The results stress on the importance of planning in a knowledge-driven generation. For tasks requiring complex reasoning, the "Plan + Gen" approach outperforms direct generation (`Gen`), indicating that structured planning significantly improves task performance.

**One example may bias the SLM.** Involving a single example in the instruction can introduce bias in Sequence Learning Models (SLMs). A specific example might cause the SLM to over-follow to certain information, leading to hallucinations. For instance, if the example includes a reference to `"singer"`, the SLM may generate plans that include `"singer"` even when the question pertains to an unrelated topic, such as `"cars"`. This observation highlights the lack of robustness in SLMs when exposed to overly specific examples. Consequently, it is better to provide more general, transferable knowledge in instructions. The degraded performance observed with 1 RAG example supports this conclusion, as the model becomes overly reliant on the provided information.

**More examples do not always improve performance.** Interestingly, increasing the number of RAG examples (from 1 to 5) results in a performance drop. This suggests that longer input sequences may confuse the SLM, making it more difficult to distill relevant knowledge. Based on these findings, we recommend using 3 RAG examples as the optimal balance for complex DSCG tasks since it avoids both the biases of a single example and the confusion caused by too many examples.

| Embedding Model | RAG Index | Reasoning Type | Instruction Type | # RAG Examples | Performance |
|---|---|---|---|---|---|
| code-t5+ | question | plan + gen | no examples | 3 | 33.80 |
| code-t5+ | question | gen | no examples | 3 | 31.40 ($\downarrow$2.40) |
| bge-large | question | plan + gen | no examples | 3 | 30.00 ($\downarrow$3.80) |
| code-t5+ | question | plan + gen | w/ examples | 3 | 28.00 ($\downarrow$5.80) |
| code-t5+ | question+schema | plan + gen | no examples | 3 | 32.40 ($\downarrow$1.40) |
| code-t5+ | question | plan + gen | no examples | 5 | 31.80 ($\downarrow$2.00) |
| code-t5+ | question | plan + gen | no examples | 1 | 29.80 ($\downarrow$4.00) |

Table 5: Ablation Study Results of AHA-MI of Phi-3-mini on BIRD-SQL. The table compares different embedding models, RAG index (with or without schema), reasoning approaches (planning or direct generation), and varying numbers of RAG examples.

# F  ERROR ANALYSIS

We conducted an error analysis by sampling 50 incorrect cases for both AHA-MI and AHA-GI across three datasets. Although AHA substantially improves the overall performance of SLMs, we found that 54% of the errors were caused by over-reasoning. This issue tends to emerge even in relatively simple cases. As discussed earlier, SLMs can overly adhere to the instructions derived from planning and guidance, which is problematic when the task is enough simple and does not require decomposition or reasoning. In these cases, direct code generation would lead to more accurate results. The remaining errors stem from common issues in code generation tasks, such as incorrect string handling, incorrect column selection, database constrain understanding.

# G  LIMITATIONS AND FUTURE WORK

A key limitation of our current approach with AHA is the reliance on initial training examples for both LLMs and SLMs to facilitate orchestration. This is why we selected datasets that include a training corpus suitable for distilling knowledge. However, an important avenue for future work is to explore how to generate such training data in a fully zero-shot manner, without relying on human-annotated or enumerated examples. Additionally, as highlighted in the error analysis, over-reasoning negatively impacts performance on simpler tasks, where additional reasoning or decomposition is unnecessary. To address this, future work could focus on developing or prompting smaller models to act as routers, as proposed by Ding et al. (2024), to classify questions based on whether they require planning. This would help avoid over-reasoning in straightforward cases and improve the overall efficiency of AHA.

## H  MAIN PROMPTS

The zero-shot End-to-End Code Generation prompt is shown in Figure 7, Figure 15 and 17 show the zero-shot Chain-Of-Thought reasoning. Figure 18 shows few-shot demonstration prompting. The `few_shot_examples` can be selected by human experts as Static Few-Shot Demonstration, and can be retrieved from AHA memory database by RAG system as Dynamic Few-Shot Demonstration.

The Figure 7, 8, 9, 10 show prompts for Orchestration between LLMs and SLMs. Figure 11 presents how LLM convert orchestrated successful cases to more understandable case studies to SLMs. LLMs can go through correct cases from memory databases and distill knowledge to an offline and plug-and-plan General Instruction for SLMs to used for new and unseen queries performed by prompts shown in Figure 12 and 13. During inference, SLMs can produce Meta Instructions by prompts in Figure 14. Given distilled knowledge (instructions), SLMs will plan first as shown in Figure 16, and generate codes finally with their knowledge-driven planning, which shows in Figure 17.

## I  KNOWLEDGE DISTILLATION EXAMPLES

### I.1  CASE STUDY EXAMPLE

The Figure 19 shows the example of case studies on Python task. The Figure 20 and Figure 21 present examples of AHA-GI and AHA-MI respectively.

```
You are a data analyst. Given the data, you need to generate the code first to answer the question:

# Please Follow:
- Do not add data inspection in the plan, such as `df.head()` or `print(df.head())` since this is
cheating!
- Do not use any external information.
- The code should be end-to-end, so you cannot encourage yourself to print other things except
  final result. More other information lead to be distracted.

# Question: {question}
# Thought: I need to see the data samples in the first 10 rows:

# Code:
```python
import pandas as pd
df = pd.read_csv('{data_path}', sep='\t')
print(df.head(10))
```

# Observation:
{data_overview}

# Thought: I can generate remaining code to answer this question:
# Code:
```python
import pandas as pd
```

Figure 7: Prompt of baseline end-to-end generation for tasks requiring Python.

```
You are a data analyst trainer. You are educating your student to generate right code to answer
tabular data analysis questions. In order to do so, you need to convert your code to code_plan and
let your students to fill to understand plans and analysis. So you cannot generate code by your
own, only generate plans.

# Data Overview at the path {data_path} (first ten rows):
{data_overview}
...

# Question: {question}
# Original Code:
```python
{ground_truth code}
```

You should convert the code into the code_plan format with the placeholder `[FILL YOUR CODE HERE]`:
```code_plan
import ...

# Step 1:....
[Fill Your Code]

# Step 2:....
[Fill Your Code]
...

# Step N: ....
```

Generate your code_plan for your student. DO NOT generate any code by your own. Also ignore and
remove steps of inspecting the data which leads to student cheating.
Please note it's hard for your student to write long code. You will get 1,000 dollars if you have
a good job:
```

Figure 8: Prompt converting ground-truth code to functional plan for python task as example. This is conducted by **LLM Teacher Agent**.

```
You are a data analyst. Given the data, expert customized functional plan, complete
each line of code to answer questions correctly:

# Please Follow:
- Do not add data inspection in the plan, such as `df.head()` or `print(df.head())`
  since this is cheating!
- Do not use any external information.
- The code should be end-to-end, so you cannot encourage yourself to print other
  things except final result. More other information lead to be distracted.

# Question: {question}
# Function Plan:
```python
{functional plan}
```
# Your entire completion code for function plan executable and correct:

# Code:
```python
import pandas as pd
```

Figure 9: Prompt of orchestration coding. This is conducted by **SLM Student Agent**.

```
You are an expert in error analysis and code planning. Your task is to guide your intern in filling out the code for your logic. You need to generate textual plans
as comments that include essential import statements, logics. Currently, the mixed code filled by your intern is incorrect. Then you should analyze and help him.

---------------------------------------- case begin: ----------------------------------------
{last turn case}
---------------------------------------- end ----------------------------------------

You are experienced data analysis programmer responsible for checking the errors, analyzing the reasons, and helping them correct the code. Note that you cannot
fill the code for them directly. You have four options for actions:
1. **Decomposition(Step Number, new sub steps**: If a step is too complicated and exceeds the intern's capability, decompose this step into multiple smaller steps
for them to fill step by step.
    Actually, you have to decompose steps if there are multiple functions or multiple lines of code in one step since they are not capable!
    step a -> step b, step c
2. **ALT(Step Number, what do you want to alt in details**: If a step is ambiguous or requires additional information or options, provide an alternative approach
or clarification. But this is a closed-book education, you cannot teach them to use external information aside code and data samples.
    step a -> step b
3. **ADD(Step Number, what do you want to add in details**: If the original step lacks important operations, add a supplementary step to ensure the main code logic
is smooth. But this is a closed-book education, you cannot teach them to use external information aside code and data samples.
    Also all available data are shown, you cannot add or teach them to use `df.head()` to overview data again.
    step a, step c -> step a, step b, step c
4. **DELETE(Step Number, what do you want to delete in details**: If some steps are unnecessary and hinder the intern's understanding of the overall logic, delete
them.
    step a, step b -> step b (deleted step a)
5. **SIMPLIFY(Step Number, simplify specific steps)**: If a step is implemented using recursion and this approach is too complex for the intern to understand or
debug, suggest a non-recursive approach that achieves the same result.
This might involve using iterative methods or other strategies to simplify the logic. If you find code fails due to this, simplify the functions.
step a (recursive) -> step a (iterative)
6. **SWITCH(Step Name, packages to SWITCH)**: If a function relies heavily on a specific package that is known to be complex or not beginner-friendly, suggest
switching to a more intuitive or simpler package that achieves similar functionality. This can help the intern understand the underlying logic without getting
bogged down by the complexities of the original package.
    step a (uses ComplexPackage) -> step a (uses SimplePackage)

You have to provide reasons based on analysis of errors for choosing this action and show your action in <action></action>, then. Finally, you must execute your
chosen action to change original code and fill in the following format:

# format:
Reason:
<reason>...</reason>
Act:
<action>...</action>

# Updated code plan:
```code_plan
import ...

# Step 1:....
[Fill Your Code]

# Step 2:....
[Fill Your Code]
...

# Step N: ....
```

## Please note:
- Do not ask students to add data inspection in the code, such as `df.head()` or `print(df.head())`
- Do not guide them to use any external information.
- The mixed code should be end-to-end, so you cannot encourage student to print other things except the final result. More other information would cause student to
be distracted.
- Just focus on how to make students learn how to better plan in the end-to-end code generation.

OK, now change your codes according to your actions.
If you don't follow rules, then you will lose 1 million dollars:
```

Figure 10: Prompt of plan optimization. This is conducted by **LLM Teacher Agent**.

```
You are a data analyst trainer. You are educating your student to generate pythyon code to answer tabular data analysis questions.

This is a successful case of your code, perform a case study on this:
---------------------------------------- case begin: ----------------------------------------
# Question: {question}

# Data Overview at the path {data_path} (first 10 rows):
{data_overview}
...

# Code:
```python
{final orchestrated code}
```
---------------------------------------- case end: ----------------------------------------

perform a concise case study! Your case study should only contain

### Case Study: [Case Name]
### Question: [Question]
### Table Info: [Summarized Useful information about Tabular Data]
### Objective:
### Explanation:

Please note your case study should make your student understand. You don't have to include code again. You will get 1000 dollars if
you have a good job:
```

Figure 11: Prompt of case study conversion. This is conducted by **LLM Teacher Agent**.

```
You are a data analyst trainer. You are educating your student to generate correct python pandas code to answer tabular
data analysis questions. To test and elicit their knowledge of python pandas code, you generate step-by-step plans that
allow them to fill in code until they succeed.

# These are case studies where they fill the correct code:
---------------------------------------- case begin: ----------------------------------------
{case_study_batch}
---------------------------------------- case end: ----------------------------------------

# Please note:
- Do not ask students to add data inspection in the code, such as `df.head()` or `print(df.head())`
- Do not guide them to use any external information.
- The mixed code should be end-to-end, so you cannot encourage student to print other things except the final result. More
other information would cause student to be distracted.
- Just focus on how to make students learn how to better plan in the end-to-end code generation.

According to the previous case studies, analyze and reflect how to generate plans which can make your student fill the
correct code. Summarize 5-7 key points.
```

Figure 12: Prompt of aggregation prompt of each batch of case studies. This is conducted by **LLM Teacher Agent**.

```
You are a data analyst trainer. You are educating your student to generate correct python pandas code to answer tabular
data analysis questions. To test and elicit their knowledge of code, you generate step-by-step plans that allow them to
fill in subqueries until they succeed.

# These are case studies where they fill the correct code:
----------------------------------------- case begin: -----------------------------------------
{last layer of case studies}
----------------------------------------- case end: -----------------------------------------

# Please note:
- Do not ask students to add data inspection in the code, such as `df.head()` or `print(df.head())`
- Do not guide them to use any external information.
- The mixed code should be end-to-end, so you cannot encourage student to print other things except the final result. More
other information would cause student to be distracted.
- Just focus on how to make students learn how to better plan in the end-to-end code generation.

Following case studies, please summarize 5-7 key points about how to plan and generate correct code to answer the tabular
data analysis questions accurately.
Students will take your notes directly.

# You need to start with:
```1. You should
```

Figure 13: Prompt of summarization prompt of batch of case studies in the last layer. This is conducted by **LLM Teacher Agent**.

```
You are a data analysis trainer. Your are teaching your student to plan and generate python code
accurately. You find some case study for reference.

# There are case studies:
----------------------------------------- case begin: -----------------------------------------
{case_studies}
----------------------------------------- case end: -----------------------------------------

Following case studies, please summarize key 5-7 points about how to plan and generate correct python
code to answer the data analysis questions accurately.

# You will use them to educate your student:
```successful plan suggestions:
1. You Should
```

Figure 14: Prompt of in-time summarization for meta-instructions. This is conducted by **SLM Student Agent**.

```
You are a data engineer. Given the sample data, generate python code plan to answer the question
accurately.

# Please Follow:
- Do not add data inspection in the plan, such as `df.head()` or `print(df.head())` since this is
cheating!
- Do not use any external information.
- The code should be end-to-end, so you cannot encourage yourself to print other things except final
  result. More other information lead to be distracted.
```

# Question: {question}
# Thought: I need to see the data samples in the first 10 rows:

# Code:
```python
import pandas as pd
df = pd.read_csv('{data_path}', sep='\t')
print(df.head(10))
```

# Observation:
{data_overview}

# Thought: I should have a step-by-step text plan for generating this code first. I will fill my plan
into the template in details:
```code_plan
Step 1: ...
Step 2: ...
...
Final Step: ...
```

Generate your plan step by step for the question:

# Let's think step by step:
```code_plan
Step 1:
```

Figure 15: Prompt of generating Chain-Of-Thought. This is conducted by **SLM Student Agent**.

```
You are a data engineer. Given the sample data, generate python code plan to answer the question

# Please Follow:
- Do not add data inspection in the plan, such as `df.head()` or `print(df.head())` since this is
  cheating!
- Do not use any external information.
- The code should be end-to-end, so you cannot encourage yourself to print other things except final
  result. More other information lead to be distracted.

# There are some important successful plan suggestions from experts:

```successful plan suggestions:
{successful_plan_suggestions}
```

# Question: {question}
# Thought: I need to see the data samples in the first 10 rows:

# Code:
```python
import pandas as pd
df = pd.read_csv('{data_path}', sep='\t')
print(df.head(10))
```

# Observation:
{data_overview}

# Thought: Referring to [successful plan suggestions], I should have a step-by-step text plan for
generating this code first. I will fill my plan into the template in details:
```code_plan
Step 1: ...
Step 2: ...
...
Final Step: ...
```

Generate your plan step by step for the question:

# Let's think step by step:
```code_plan
Step 1:
```

Figure 16: Prompt of knowledge-driven planning. This is conducted by **SLM Student Agent**.

```
You are a data engineer. Given the sample data, generate python code to answer the question accurately.

# Question: {question}
# Thought: I need to see the data samples in the first 10 rows:

# Code:
```python
import pandas as pd
df = pd.read_csv('{data_path}', sep='\t')
print(df.head(10))
```

# Observation:
{data_overview}

# Thought: I can generate code to answer this question and print the result. I will fill my code in the
template:
```python
[Your Code]
```

Let's think step by step for the question:
{step-wise plans}

# Code:
```python
import pandas as pd
```

Figure 17: Prompt of code generation given step-wise planning. This is conducted by **SLM Student Agent**.

```
You are a data analyst. Given data sample, you need to generate pandas code first to answer the question.

Generate your pandas code to answer the question, and print the result for your to understand. Fill your
code in
```python
[Your Code]
```

# Please follow:
- Do not add data inspection in the plan, such as `df.head()` or `print(df.head())` since this is
cheating!
- Do not use any external information.
- The code should be end-to-end, so you cannot encourage yourself to print other things except the final
result. More other information would cause sutdent to be distracted.

There are some examples:
-------------------------- Examples Start --------------------------
{few_shot_examples}
-------------------------- Examples END --------------------------

# Question: {question}
# Thought: I need to see the data samples in the first 10 rows:

# Code:
```python
import pandas as pd
df = pd.read_csv('{data_path}', sep='\t')
print(df.head(10))
```

# Observation:
{data_overview}

# Thought: I can generate code to answer this question:

# Code:
```python
import pandas as pd
```

Figure 18: Prompt of few-shot demonstration. This is conducted by **SLM Student Agent**.

```
### Case Study: Average Weight Calculation for Specific Players

### Question:
What is the average weight of Jamarr Sanders and Robert Williams?

### Table Info:
- **Columns**: Name, Height, Weight (lbs.), Position, Class, Hometown, Previous Team(s)
- **Sample Data**:
  - Jamarr Sanders: Weight 210 lbs.
  - Robert Williams: Weight 210 lbs.

### Objective:
To calculate the average weight of the players Jamarr Sanders and Robert Williams from the given dataset.

### Explanation:
1. **Load Data**: The data is loaded from a tab-separated values (TSV) file.
2. **Filter Data**: Rows corresponding to the names "Jamarr Sanders" and "Robert Williams" are filtered from the dataset.
3. **Calculate Average**: The average weight of the filtered rows is computed.
4. **Output**: The result is printed as an integer.

By following these steps, the student can understand how to filter specific rows in a dataset and perform calculations on the filtered data. This case
demonstrates the practical application of data manipulation and analysis using pandas in Python.
```

Figure 19: Example of case studies for tasks requiring Python. This is conducted by **LLM Teacher Agent**.

1. You should **break down the task into manageable steps**. Each step should build on the previous one, guiding you through the process logically.

2. You should **emphasize data handling and cleaning**. This includes handling missing values, normalizing case, and ensuring data consistency."

3. You should **focus on filtering and extraction**. Guide yourself on how to filter and extract relevant data based on specific criteria. This is often the core of the analysis."

4. You should **perform aggregation and counting**. Learn how to perform aggregation operations like counting, summing, or finding minimum/maximum values to derive insights from the data."

5. You should **present the result clearly**. Ensure that the final step involves presenting the result in a clear and concise manner. This reinforces the importance of communicating findings effectively."

6. You should **avoid distractions**. Keep the instructions focused on the end-to-end process without encouraging unnecessary intermediate outputs or external information. This helps maintain your focus on the task at hand."""

Figure 20: Example of General Instruction for tasks requiring Python. This is conducted by **LLM Teacher Agent**.

"question": "which country rank last?"

1. Understand the problem statement and the data structure.

2. Load the data using appropriate libraries (e.g., pandas).

3. Perform necessary data manipulation and cleaning.

4. Identify the relevant columns and values for the analysis.

5. Use appropriate functions and methods to filter, sort, and extract the required information.

6. Output the result in a clear and concise manner.

Figure 21: Example of General Instruction for tasks requiring Python. This is conducted by **SLM Student Agent** in time.

