# OpenReview forum: "Agents Help Agents: Exploring Training-Free Knowledge Distillation for Small Language Models in Data Science Code Generation"
_ICLR.cc/2025/Conference — Submitted to ICLR 2025_

### Official Review · Reviewer_3CZZ · 2024-11-04

**Soundness:** 2
**Presentation:** 3
**Contribution:** 2
**Rating:** 5
**Confidence:** 3

**Summary:**

Agents Help Agents is a framework in which large language models (LLMs) assist with providing Small Language Models (SLMs) with the necessary in-context information for solving data science code generation problems.

This framework has three phases. In the first phase, known as the Agent Orchestration Interface (AOI), the LLM is required to generate a step-by-step plan. From this, the SLM is required to complete the code based on the plan. The ground-truth code is provided, and so the LLM will update the substeps in the plan accordingly until the SLM correctly solves the problem or after a finite number of steps.

In the second phase, a memory database is constructed. For each successfully solved problem by the SLM, the LLM provides a case study explaining why the problem was successfully solved.

In the third phase, a general instruction or retrieval-augmented instruction generation can be sent to the SLM. This is then used as part of the prompt as distilled in-context knowledge which can be used for solving a query in the test set.

The general instruction and the retrieval augmented instruction approach are then both benchmarked against CoT, few-shot, and larger SLMs than Phi-3-Mini. From an aggregated standpoint of the table, the AHA method is better than the other baselines.

**Strengths:**

1.This is a novel framework that has clear benefits when showing that SLMs can exceed chain-of-thought, few-shot prompting, and larger SLMs in data science code generation. Another strength is that the authors successfully demonstrated that the approach is model agnostic.

2.The task decomposition approach is novel and highly effective. In allowing the teacher agent to decompose its plan down and iteratively refine it for the SLM, this allows for effective case studies for the memory bank. Section 3.6 and Figure 2 also further highlight the efficacy of this method when showing how the plan decomposition is superior to CoT as the number of turns increases.

3.The method is thoroughly tested with three different DSCG frameworks, thereby clearly showing the efficacy of this framework for this specific task.

**Weaknesses:**

One major weakness with this paper is that the broader impact of this approach is rather limited. I believe that this method could be applicable to several other use cases beyond just data science code generation. While the privacy-preserving component makes it particularly relevant to DSCG, I think that it is also useful for several use cases where it is cheaper to run SLMs agents for task-solving.

Additionally, there is minimal benchmarking against other student-teacher methods which have been used for instruction optimization such as ADAS (https://arxiv.org/pdf/2408.08435), DSPy (https://arxiv.org/pdf/2310.03714) and TextGrad (https://arxiv.org/pdf/2406.07496). These should also be mentioned in the Related Works section since they follow a similar paradigm.

Lastly, I feel that an ablation study should also be performed to show the necessity of each step as the current framework. The current AHA framework seems somewhat complicated with many steps. Hence, it would be good to see if all the steps are necessary, along with which steps have the most impact and their corresponding hyperparameters.

**Questions:**

1.Instead of using an LLM as the teacher agent, I was wondering what the performance would be like if a SLM such as Phi-3-mini was used for the knowledge distillation? This would be helpful in knowing the role that this structure plays.
2.How does this method compare to other student-teacher methods such as DSPy, TextGrad, and ADAS where a teacher model works on helping a student model to optimize general instructions?
3.For Section 3.6 and Figure 4a, can you please clarify how you did additional turns of CoT? Was the LLM teacher agent attempting to improve the instructions for CoT each time? If not, did the SLM just iteratively adjust its explanation over the epochs?

---

### Official Review · Reviewer_NTFu · 2024-11-04

**Soundness:** 2
**Presentation:** 3
**Contribution:** 1
**Rating:** 3
**Confidence:** 3

**Summary:**

This paper presents a pipeline for small language models on data science code generation due to the privacy and cost-related concerns of using larger language models. To help smaller models, authors apply orchestrated code generation on seedling data to induce higher-quality examples from larger language models and collect them as a knowledge base. During inference, the pipeline samples examples to apply in-context learning on the small language model. Experiments verify improved scores on small language models on DSCG.

**Strengths:**

1. a novel framework for using large language models to help smaller language models on DSCG via in-context learning.
2. experiments verify more or less improved code generation results.

**Weaknesses:**

1. Misleading terms used. Defaulting in-context learning as a sub-category of knowledge distillation is still less used in the field, with the closest work being an in-context learning distillation paper. Using more rigorous terms in the title will be more appropriate. Similarly, the usage of agents in this paper also needs more explanation, as the "actions" in this scenario are quite limited to code generation (especially considering that agents are writing codes collectively), which is the default function of language models. Using such terms in the paper body and title can be highly misleading.

2. The claim made in the paper lacks Sufficient experimental support. The paper's reliance on a single small language model for evaluation is a limitation. Given the complexity of the pipeline, the absence of a comprehensive ablation study to validate each step's efficacy is another significant gap.

3. There are mysterious results that many in-context learning baselines, such as static few-shot and chain-of-thought, underperform the backbone in many items. These are likely due to the very large variance of the results or the uncareful design of the baselines. More explanations of such phenomena would be needed to support the reliability of the claims.

4. The experiments are implemented on DSCG which is a limited domain considering that the pipeline should have been generally applicable across wider areas, which limits the contribution and impact of the work.



Huang, Yukun, Yanda Chen, Zhou Yu, and Kathleen McKeown. "In-context learning distillation: Transferring few-shot learning ability of pre-trained language models." arXiv preprint arXiv:2212.10670 (2022).

**Questions:**

1. typo: GPT-35-turbo -> GPT-3.5-turbo

---

### Official Review · Reviewer_XVWX · 2024-11-04

**Soundness:** 3
**Presentation:** 2
**Contribution:** 3
**Rating:** 6
**Confidence:** 3

**Summary:**

This paper proposed a training-free knowledge distillation method for data science code generation. It claims that previous methods based on fine-tuning are resource-consuming. The method is composed of three steps -- First, the teacher model generates coarse code blocks (plans), and the student model fills the detailed codes. Second, only the high-quality codes are kept and sent to a _memory database_. During knowledge distillation, a dynamic RAG component is used to raise the performance on long-tail examples.

**Strengths:**

- The motivation of the study is clear. Fine-tuning-based knowledge distillation requires a decent amount of computation resources, so a workaround that's more efficient is desirable.
- The organization and writing of the paper are quite good. Overall I enjoyed reading the paper.

**Weaknesses:**

- When the ground truth code is generated by the Teacher Agent, there could be mistakes / errors / inaccuracies. When the gt code is given as a condition to generate the functional plan, there could be error propagation - the generated functional plan has error too?
- Some details are not quite clear. In Section 2.4, the details of constructing the knowledge tree is missing.
- More references could be potentially included, commented and compared, such as CodeAct [1]


Minor issues:
Line 26: GPT-35-turbo -> GPT-3.5-turbo
Line 124: where L_al LLM performing ... -> where L_al means LLM performing...

References:
[1] Executable Code Actions Elicit Better LLM Agents

**Questions:**

- What does the "ground-truth answer string" mean? The execution output of the code?
- I do not quite understand the motivation of constructing a knowledge tree during the knowledge distillation phase. Could you further elaborate that?

---

> ### Comment · Reviewer_XVWX · 2024-12-03
> **Reviewer Response**
>
> I thank the authors for clarifying my confusions. I have revised my scores to reflect my new evaluation of this paper.

---

### Official Review · Reviewer_s3LZ · 2024-11-06

**Soundness:** 2
**Presentation:** 2
**Contribution:** 2
**Rating:** 3
**Confidence:** 5

**Summary:**

This paper has reduced margins on both the left and right sides to fit more content into 10 pages, which is a Margin violation.

**Strengths:**

This paper has reduced margins on both the left and right sides to fit more content into 10 pages.

**Weaknesses:**

This paper has reduced margins on both the left and right sides to fit more content into 10 pages, which is a Margin violation.

**Questions:**

This paper has reduced margins on both the left and right sides to fit more content into 10 pages, which is a Margin violation.

---

### Meta-Review · Area_Chair_yDeZ · 2024-12-19

**Metareview:**

The paper "Agents Help Agents: Exploring Training-Free Knowledge Distillation for Small Language Models in Data Science Code Generation" proposes a novel framework, Agents Help Agents (AHA), for training-free knowledge distillation from large language models (LLMs) to small language models (SLMs) in the context of data science code generation (DSCG). AHA leverages In-Context Learning (ICL) to facilitate automatic knowledge transfer via an agent orchestration interface (AOI), memory collection, and inference augmentation with distilled knowledge. The framework is evaluated on three challenging DSCG tasks: TabMWP, BirD-SQL, and WikiTQ, demonstrating significant performance improvements for SLMs, notably Phi-3-mini, Llama-3.1-8B, and GPT-3.5-Turbo. The paper also provides insights into the model-agnostic nature of the distilled knowledge and compares different distillation strategies.

#### Contributions
The primary contributions of the paper are:
1. **AHA Framework**: A novel training-free knowledge distillation framework using ICL, enabling SLMs to benefit from LLMs without resource-intensive training.
2. **Performance Improvements**: Demonstrates significant relative performance improvements (e.g., 27.5% for Phi-3-mini, 14.3% for Llama-3.1-8B, and 30.9% for GPT-3.5-Turbo) across DSCG tasks, validating the effectiveness of AHA.
3. **Model-Agnostic Knowledge**: Shows that the distilled knowledge in AHA is applicable to SLMs not involved in the orchestration process, highlighting its generalizability.
4. **Insights into DSCG Challenges**: Provides detailed analysis and insights into the unique challenges of DSCG, such as handling heterogeneous inputs and the limitations of traditional few-shot prompting in this domain.



#### Weaknesses
1. **Formatting Issues**: The paper initially included a formatting error due to the use of a `geometry` package, leading to reduced margins. This was corrected, and the paper now adheres to the 10-page limit, but the initial oversight caused one reviewer (s3LZ) to refuse reviewing it.

2. **Clarity and Terminology**: Some reviewers (NTFu, 3CZZ) noted that the use of terms like "knowledge distillation" and "agents" could be misleading or overly broad, as the paper primarily focuses on ICL rather than traditional distillation methods, and the agent actions are limited to code generation.

3. **Scope and Generalizability**: Concerns were raised about the paper's focus on DSCG, which is seen as a narrow domain, potentially limiting the broader impact of the method (NTFu, 3CZZ). While the authors argue for the complexity and real-world relevance of DSCG, the lack of experimentation in other domains was noted.

4. **Experimental Rigor**: The absence of a comprehensive ablation study was highlighted by multiple reviewers (NTFu, 3CZZ), suggesting that the necessity and impact of each step in the AHA framework should be more thoroughly validated. Additionally, the reliance on a single SLM (Phi-3-mini) for primary evaluation was seen as a limitation (NTFu).

5. **Comparative Analysis**: The paper lacks comparisons with other student-teacher methods, such as ADAS, DSPy, and TextGrad, which follow similar paradigms (3CZZ). While the authors later conducted a comparison with DSPy, this was not part of the initial submission.

6. **Unexplained Results**: Some in-context learning baselines (e.g., static few-shot, chain-of-thought) underperformed the backbone model, which was attributed to high variance or baseline design issues. More explanation was deemed necessary to support the reliability of the claims (NTFu).

7. **Reviewer Communication**: The authors expressed frustration with a reviewer (3CZZ) who adjusted their score without providing new feedback, despite the authors' efforts to address initial concerns. This lack of transparency in score adjustments added to the authors' concerns about the review process.

**Additional Comments On Reviewer Discussion:**

1. **Formatting Error (s3LZ):**
   - **Concern**: The initial formatting error led to a margin violation, prompting one reviewer to refuse reviewing the paper.
   - **Response**: The authors corrected the formatting issue, reducing table sizes and adhering to the 10-page limit, and sought guidance on whether they could still participate in the discussion phase.
   - **Impact**: The issue was resolved, but the initial refusal by the reviewer impacted the overall review process.

2. **Terminology and Scope (NTFu, 3CZZ):**
   - **Concern**: The use of "knowledge distillation" and "agents" was deemed potentially misleading, and the focus on DSCG was seen as limiting the broader impact.
   - **Response**: The authors clarified that they did not claim ICL as a subcategory of knowledge distillation but as an alternative mechanism for knowledge transfer. They also emphasized the complexity and real-world significance of DSCG, citing prior ICLR papers in the domain.
   - **Impact**: The clarifications addressed some concerns, but reviewers maintained their views on the scope and terminology.

3. **Experimental Rigor and Ablation Studies (NTFu, 3CZZ):**
   - **Concern**: The lack of comprehensive ablation studies was highlighted, along with the reliance on a single SLM for primary evaluation.
   - **Response**: The authors provided additional ablation studies for the exploration phase and emphasized that the comparison between zero-shot baselines and AHA-GI served as an effective ablation for the inference phase. They also conducted experiments with additional SLMs to demonstrate generalizability.
   - **Impact**: The additional experiments and clarifications partially addressed concerns, though reviewers still emphasized the need for more rigorous validation.

4. **Comparative Analysis (3CZZ):**
   - **Concern**: The absence of comparisons with other student-teacher methods like ADAS, DSPy, and TextGrad was noted.
   - **Response**: The authors conducted a post-hoc comparison with DSPy, highlighting the limitations of DSPy in DSCG tasks and emphasizing the unique focus of AHA on task-specific instructional guidance.
   - **Impact**: The comparison addressed the concern, but the lack of initial inclusion was noted by the reviewer.

5. **Unexplained Results (NTFu):**
   - **Concern**: The underperformance of some in-context learning baselines was questioned, suggesting potential issues with variance or baseline design.
   - **Response**: The authors provided detailed explanations, referencing prior studies and discussing the complexity of data structures in DSCG tasks.
   - **Impact**: The explanations were acknowledged but did not lead to a change in the reviewer's score.

**Final Decision Weighting:**
The paper presents a novel and impactful framework for training-free knowledge distillation in DSCG, demonstrating significant performance improvements and model-agnostic knowledge transfer. The authors' responsiveness to reviewer feedback, including additional experiments and clarifications, highlights their commitment to rigor. However, concerns about the scope, terminology, and experimental validation persist. The lack of initial comparisons with relevant works and the formatting issue also impacted the review process.

---

### Decision · Program_Chairs · 2025-01-22

Reject